# Autoimmune Encephalitis and CSF Anti-GluR3 Antibodies in an MS Patient after Alemtuzumab Treatment

**DOI:** 10.3390/brainsci9110299

**Published:** 2019-10-30

**Authors:** Maria Chiara Buscarinu, Arianna Fornasiero, Giulia Pellicciari, Roberta Reniè, Anna Chiara Landi, Alessandro Bozzao, Cristina Cappelletti, Pia Bernasconi, Giovanni Ristori, Marco Salvetti

**Affiliations:** 1Department of Neurosciences, Mental Health and Sensory Organs, Centre for Experimental Neurological Therapies (CENTERS), Faculty of Medicine and Psychology, Sapienza University, 00189 Rome, Italy; mchiara.buscarinu@gmail.com (M.C.B.); ari.fornasiero@gmail.com (A.F.); alessandro.bozzao@uniroma1.it (A.B.); 2Faculty of Medicine and Psychology, Sapienza University, 00189 Rome, Italy; giuliapellicciari11@gmail.com (G.P.); roberta.renie@gmail.com (R.R.); landiannachiara@gmail.com (A.C.L.); 3Neurology IV–Neuroimmunology and Neuromuscular Diseases Unit, Fondazione IRCCS Istituto Neurologico Carlo Besta, 20133 Milan, Italy; cristina.cappelletti@istituto-besta.it (C.C.); pia.bernasconi@istituto-besta.it (P.B.); 4IRCCS Istituto Neurologico Mediterraneo (INM) Neuromed, 86077 Pozzilli, Italy

**Keywords:** multiple sclerosis, autoimmune diseases, immune thrombocytopenic purpura, autoimmune encephalitis, alemtuzumab, antibodies against GluR3 peptide

## Abstract

A 45-year-old Italian woman, affected by relapsing–remitting multiple sclerosis (RR-MS) starting from 2011, started treatment with alemtuzumab in July 2016. Nine months after the second infusion, she had an immune thrombocytopenic purpura (ITP) with complete recovery after steroid treatment. Three months after the ITP, the patient presented with transient aphasia, cognitive deficits, and focal epilepsy. Serial brain magnetic resonance imaging showed a pattern compatible with encephalitis. Autoantibodies to glutamate receptor 3 peptide A and B were detected in cerebrospinal fluid and serum, in the absence of any other diagnostic cues. After three courses of intravenous immunoglobulin (0.4 mg/kg/day for 5 days, 1 month apart), followed by boosters (0.4 mg/kg/day) every 4–6 weeks, her neurological status improved and is currently comparable with that preceding the encephalitis. Autoimmune complications of the central nervous system during alemtuzumab therapy are relatively rare: only one previous case of autoimmune encephalitis following alemtuzumab treatment has been reported to date.

## 1. Introduction

Alemtuzumab, a humanized monoclonal antibody indicated for the treatment of patients with relapsing–remitting multiple sclerosis (RR-MS), increases the risk of autoimmune adverse events, including thyroid disorder, renal disease, and immune thrombocytopenic purpura (ITP) [1]. Recently, new complications after alemtuzumab treatment have been described, like stroke, myocardial infarction, diffuse alveolar hemorrhage, and hemophagocytic lymphohistiocytosis (as reported in the European Medicine Agency note EMEA/H/A-20/1483/C/3718/0028).

We here report a case of presumed autoimmune encephalitis (AE) after the second course of alemtuzumab. AE is one of the most common causes of non-infectious encephalitis, with a variety of clinical manifestations, including behavioral and psychiatric symptoms, autonomic disturbances, movement disorders, and seizures. First-line immune therapies in AE consist of corticosteroids (intravenous and oral), sometimes coupled with intravenous immunoglobulin (IVIG) and/or plasma exchange (PE). Second-line treatments, including rituximab, cyclophosphamide, azathioprine, and mycophenolate mofetil, are administered when the first-line therapies fail to produce adequate benefits, when the disease is severe or relapsing, or, even in case of response to first-line treatments, with the goal of decreasing the risk of relapse in AE [2].

## 2. Case Report

We report the case of a 45-year-old Italian woman affected by RR-MS from 2011, when she had a diplopia and underwent a magnetic resonance imaging (MRI) showing multiple contrast-enhancing lesions in her brain and spinal cord white matter. After a spontaneous recovery, she later had another clinical attack and was treated with high intravenous steroids. Having fulfilled the criteria of definite diseases, a spinal tap was not performed and a disease-modifying therapy was started. After the failure of two first-line therapies (glatiramer acetate and dimethylfumarate) with clinical reactivations and new lesions identified after a new MRI, she started alemtuzumab in July 2016. Other second-line treatments, including natalizumab and fingolimod, were contraindicated for the presence of anti-JC virus antibodies at high titer (stratify index 3.20) and bradycardia. The alemtuzumab schedule (12 mg once daily (QD) for 5 days, followed by 12 mg QD for 3 days after one year) was approved for MS treatment.

In June 2018, 9 months after the second alemtuzumab infusion cycle, she reported a longer and more abundant menstrual period, bleeding from the gums, and scattered red spots on the skin. She was then referred to the emergency department: her platelet level was 1000/μL (normal range: 150,000–450,000/μL), with positive direct and indirect Coombs tests, and a normal bone marrow biopsy. A diagnosis of ITP was made and steroid treatment (methyl-prednisolone 40 mg daily for 7 days, followed by tapering) was promptly started with improvement: her platelet count became normal and the symptoms regressed in approximately 30 days.

In September 2018, 3 months after ITP, the patient presented with progressive aphasia and underwent a brain MRI that showed a pattern compatible with encephalitis (Figure 1a). She was hospitalized and her neurological examinations showed a change in neurological status with anomic aphasia and motor apraxia. Cerebrospinal fluid (CSF) was clear, with a slight increase in glucose (73 mg/dl) and protein (61 mg/dl) and normal cell numbers (4 cells/mmc; normal range: 0–5 cell/mmc). Immunoelectrophocusing showed an IgG index of 1.45 (0.00–0.65) and the presence of 17 oligoclonal bands. The PCR for herpes viruses (HSV (herpes simplex virus), CMV (cytomegalovirus), VZV (varicella-zoster virus), EBV (Epstein-Barr virus), HHV6 (human herpesvirus 6)) and the JC virus (JCV) was negative. Autoimmune screening (anti-gliadin IgG e IgA, anti-transglutaminase, anti-cardiolipin, antibodies to double-stranded DNA, extractable nuclear antigens, and anti-neutrophil cytoplasmic antibodies) was negative. Serology for common and neurotropic infectious agents (Toxoplasma, B. Burgdorferi, HIV), levels of oncotumor markers (CEA (carcino-embryonic antigen), AFP (alpha fetoprotein), CA (cancer antigen) 125, CA 15-3, CA 19-9, and Cyfra (cytokeratin 19 fragment antigen) 21-1 NSE (neuron-specific enolase)), antibodies against onconeural antigens (anti-amphiphysin, anti-MA2, anti-Yo, anti-Ri, anti-Hu, anti-GAD65, anti-titin, anti-recoverin, anti-Sox1, and anti-Zic4), and a total-body computerized tomography were all normal.

Three days after hospitalization, aphasia recovered completely; however, a new brain MRI showed increased edema in the left fronto-temporal subcortical white matter, without diffusion restriction or contrast enhancement (Figure 1b). Five days after hospitalization, a focal epilepsy started with clonic movements in her left upper limb, associated with the worsening of working memory and mood changes. An electroencephalogram showed non-specific electrical alterations in the bilateral temporo-occipital lobes. The patient started therapy with oral levetiracetam (1500 mg daily), with a stop in seizures and improvement of neurological status. We chose not to start steroid or other immunomodulatory therapies, planning strict clinical and neuroradiological follow-up instead.

After twenty days, the patient presented with vomiting and mental confusion. A new MRI showed the reduction of a T2 hyperintense lesion previously described, but the appearance of five similar lesions (Figure 1c). Neurological examination showed a worsening of cognitive (especially executive) function and mood status. A second CSF examination was performed that showed 1 cell/mmc, 29 proteins mg/dl, glucose 54 mg/dl, and also included the search for a panel of autoantibodies known to be associated with AE (not investigated in the previous CSF examination). Autoantibodies to glutamate receptor 3 (GluR3) peptide A and B were detected both in the CSF (0.143 and 0.140, respectively, at CSF dilution 1:2) and in the serum (1.074 and 1.155, respectively, at serum dilution 1:200) by an enzyme-linked immunosorbent assay, as described [3,4]. No other findings emerged from the second CSF examination and we therefore started the first course of intravenous immunoglobulin (0.4 mg/kg/day for 5 days).

After a strict follow-up of about 3 weeks, during which the patient showed a partial recovery, a new MRI documented a worsening condition with an extension of signal alterations in the right frontal-orbital and temporal-basal region, with evident contrast enhancements in the hippocampus and cingulum cortex (Figure 1d). The patient underwent another two courses of intravenous immunoglobulin (0.4 mg/kg/day for 5 days), one month apart, with progressive clinical-MRI improvement (Figure 1e). Her neurological status is currently comparable with that preceding the encephalitis. Given the response to IVIG treatment, we decided to continue that treatment (0.4 g/kg/day, every 4–6 weeks), while we stopped any treatment for MS. The patient is currently on monthly follow-up visits, possibly planning B cell-depleting treatments.

## 3. Discussion

Autoimmune complications of the central nervous system (CNS) during alemtuzumab therapy are relatively rare: one case of AE was reported to date [5]. The AE case occurred seven months after the second course of alemtuzumab, presenting with a polymorphic epilepsia partialis continua/status epilepticus in a patient with previous autoimmune hypothyroidism and ITP. No autoantibodies were reported in this case.

The clinical-MRI pattern of our patient is comparable with that recently reported. However, the peculiarity of our case was the positivity for anti-GluR3 autoantibodies. This is one of the antibodies directed against ionotropic glutamate receptors. They are present in 25%–30% of patients with different types of epilepsy, underpinning forms of ‘autoimmune epilepsy’ with frequent cognitive, psychiatric, and behavioral impairments [6]. The finding of anti-GluR3 autoantibodies was also associated with Rasmussen encephalitis, where they cause complement-mediated neuronal damage, irrespective of an excitotoxic effect [7]. A recent case of intractable myoclonus associated with anti-GluR3 antibodies was reported after allogeneic bone marrow transplantation [8].

Overall, our case and that described by Giarola et al. [5] strongly suggest a relationship between AE and previous therapy with alemtuzumab. Especially in patients with other well-known alemtuzumab-associated autoimmune complications (such ITP), monitoring of clinical events that may encompass the AE spectrum is advisable.

## Figures and Tables

**Figure 1 brainsci-09-00299-f001:**
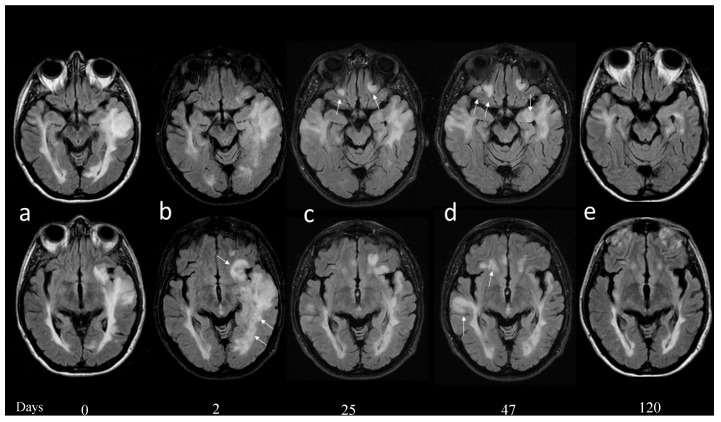
At admission (**a**) showed a large T2 signal alteration involving the left temporal lobe and expanding the superior temporal gyrus. (**b**) Two days later, a new brain magnetic resonance imaging (MRI) showed increased edema also involving the frontal subcortical and periventricular white matter (**b**, arrows); compression of the ventricular system was increased; and no diffusion restriction or contrast enhancement was demonstrated. (**c**) Twenty-five days after admission a new MRI showed reduction of the previously described T2 temporal lobe signal alteration with reduced compression of the temporal horn of the ventricle. Five new lesions were demonstrated and two of these were located in the fronto-orbital regions bilaterally (**c**, arrows). (**d**) Forty-seven days after admission, a new MRI documented a worsening of the T2 signal alterations in the fronto-orbital and in the temporal region on the right, with mild contrast enhancement in the right hippocampus and cingulum cortex (not shown). (**e**) Four months after the beginning of symptomatology all the signal alterations were markedly reduced and no enhancement was evident.

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
