# Peer review of "Autoimmune Encephalitis and CSF Anti-GluR3 Antibodies in an MS Patient after Alemtuzumab Treatment"

_brainsci, 2019, doi:10.3390/brainsci9110299_

Round 1

Reviewer 1 Report

There are extensive grammatical and spelling errors throughout the manuscript  What was the initial presentation of the original MS presentation? Did the patient have OCG at that point? What was their JCV titre What did their initial MRI show (the one that was associated with the MS diagnosis0? CSF in Sept 2018 - was the total cells 4 or subsets? What do they mean by neutrotrophic infections? Which ones were actually tested - name them please Which 'oncotumour' markers were actually tested? How did epilepsy partialis present? What did the EEG show? In terms of cognition - what was the actual deficit?

Author Response

The multipal sclerosis began with diplopia in the lateral gaze to the right; Lumbar puncture was not performed.

In October 2013 the stratify index was 3.203.

In 2012 he performed the first MRI that showed sovratentorial and undertentorial areas, an active lesion in withe matetr and areas C3, C5-C6 and C7

CSF in Sept 2018 showed normal cells count ( 4 cell/mmc, with range 0-5 cell/mmc)

Neurotrophic infection means: HSV, CMV, VZV, EBV, HHV6.

Oncotumor markers that were tested: CEA, AFP, CA 125, CA 15-3, CA 19-9, Cyfra 21-1 NSE

Partialis epilepsia occurred with clonus in left upper limb; the EEG showed modest, non-specific electrical alterations in bilateral temporo-occipital area

Actually she present slight ideomotor slowing

Reviewer 2 Report

While this is an interesting case of autoimmune encephalitis, presumably related to alemtuzumab treatment in a patient with MS, it is difficult to follow often related to typographical and grammatical errors. Improvement of these and other revisions are suggested.

Correct typographical and grammatical errors throughout manuscript that make it difficult to follow. It seems there may have been translation issues.

Abstract: Define all acronyms and ensure they are correctly listed. Ex. IPT is listed instead of ITP and this is not defined. Please make the last sentence more clear when discussing a case reported to date. Please clarify that there was 1 previous case reported and you are not referring to the current case.

Introduction: Please clarify what this refers to: (EMEA/H/A-20/1483/C/3718/0028).

Introduction: Sometimes second line treatments like rituximab are given even with response to first line therapy with a goal of decreasing risk of relapse in autoimmune encephalitis. This should be clarified and citations about this approach added.

Please include the dose and timing of alemtuzumab in the case report.

Please specify which DMT were contraindicated due to JCV Ab and cardiac issues.

Please clarify what this is: “osteomidollar biopsy”. Is this a bone marrow biopsy?

What dose and duration of steroids were used for ITP?

What is “Link Index”? Is this IgG index?

Which infectious agents were tested for? Was HSV PCR negative? Was the patient initially treated with acyclovir for possible viral encephalitis? If not, please clarify why not.

Which antibodies and autoimmune screening tests were done initially? Please clarify specific tests done in lines 58-61. Was the GluR3 Ab tested at this time?

Figure: Please label the timing of MRIs on the figure itself so it is more easy to follow. Did atrophy develop over time?

Why was the patient not empirically treated for possible AE after negative infectious work up? Could this have led to more refractory disease due to immunotherapy delay?

Please discuss the significance of antibodies to GluR3 and prior data on clinical associations with these antibodies in greater detail with additional references.

Was use of steroids or PLEX considered? Was rituximab considered?

In the discussion, it should be made more clear when describing the previously reported AE case compared to the current case. Did the prior care report identify an associated antibody?

Please further discuss clinical implications of these findings. How will the patient be treated moving forward for MS and AE

Author Response

Q: While this is an interesting case of autoimmune encephalitis, presumably related to alemtuzumab treatment in a patient with MS, it is difficult to follow often related to typographical and grammatical errors. Improvement of these and other revisions are suggested.

A: We thank the Reviewer for his expression of interest for our paper. We corrected the manuscript according to the Reviewer’s suggestions.

Q: Correct typographical and grammatical errors throughout manuscript that make it difficult to follow. It seems there may have been translation issues.

A: We corrected the errors reported by the Reviewer throughout the manuscript.

Q: Abstract: Define all acronyms and ensure they are correctly listed. Ex. IPT is listed instead of ITP and this is not defined. Please make the last sentence more clear when discussing a case reported to date. Please clarify that there was 1 previous case reported and you are not referring to the current case.

A: The abstract was rephrased.

Q: Introduction: Please clarify what this refers to: (EMEA/H/A-20/1483/C/3718/0028).

A: Done

Q: Introduction: Sometimes second line treatments like rituximab are given even with response to first line therapy with a goal of decreasing risk of relapse in autoimmune encephalitis. This should be clarified and citations about this approach added.

A: We modified the text on page 1 as follows: ‘Second-line treatments, including rituximab, cyclophosphamide, azathioprine, and mycophenolate mofetil, are administered when the first-line therapies fail to produce adequate benefit, when the disease is severe or relapsing [2], or even in case of response to first-line treatments with the goal of decreasing the risk of relapse in AE [3]’.

Q: Please include the dose and timing of alemtuzumab in the case report.

A: Done

Q: Please specify which DMT were contraindicated due to JCV Ab and cardiac issues.

A: Done

Q: Please clarify what this is: “osteomidollar biopsy”. Is this a bone marrow biopsy?

A: We replaced the term with ‘bone marrow biopsy’

Q: What dose and duration of steroids were used for ITP?

A: Done

Q: What is “Link Index”? Is this IgG index?

A: We replaced the definition with “IgG index”.

Q: Which infectious agents were tested for? Was HSV PCR negative? Was the patient initially treated with acyclovir for possible viral encephalitis? If not, please clarify why not.

A: We tested for all neurotropic herpes viruses and got negative results (page 2, lines 74, 77, 78).

Q: Which antibodies and autoimmune screening tests were done initially? Please clarify specific tests done in lines 58-61.

A: Done

Q: Was the GluR3 Ab tested at this time?

A. The GluR3 Ab was tested at the time when the patient developed vomiting and mental confusion, as reported on page 3.

Q: Figure: Please label the timing of MRIs on the figure itself so it is more easy to follow. Did atrophy develop over time?

A: We labeled the brain MRI scans in the Figure according to the days from symptom onset. No signs of atrophy developed during follow-up.

Q: Why was the patient not empirically treated for possible AE after negative infectious work up? Could this have led to more refractory disease due to immunotherapy delay?

A: The complete spontaneous remission within 3 days suggest no therapy at that time. The disease was not refractory in any case, since the response to intravenous Ig may be considered good.

Q: Please discuss the significance of antibodies to GluR3 and prior data on clinical associations with these antibodies in greater detail with additional references.

A:A new reference was added in the revised version of the case report.

Q: Was use of steroids or PLEX considered? Was rituximab considered?

A: The good response to intravenous Ig did not require alternative therapy. Concerning B cell-depleting therapy see last point.

Q: In the discussion, it should be made more clear when describing the previously reported AE case compared to the current case. Did the prior care report identify an associated antibody?

A: No associated antibody was reported (page 4, line 124).

Q: Please further discuss clinical implications of these findings. How will the patient be treated moving forward for MS and AE.

A: We added the following sentence at the end of the case report: ‘Given the response to IVIG treatment, we decided to continue that treatment (0.4 g/Kg/day, every 4-6 weeks), while we stopped any treatment for MS. The patient is currently on monthly follow-up visits, possibly planning B cell-depleting treatments’.

Round 2

Reviewer 1 Report

comments have been addressed  

Author Response

English language and style are fine/minor spell check required

Done.

Reviewer 2 Report

While the paper is improved by the revisions, there remain significant  grammatical issues that interfere with the paper's readability. The authors should consider having the paper proofread by a native English speaker.

The abstract  says the patient had “transient” aphasia, while the body of the case report states the patient had “progressive” aphasia. These are different and it should be clarified whether aphasia was transient or progressive.

Please specify in line 78-79 antibodies to onconeural antigens that were tested and whether this was on serum or CSF. Please clarify that the antibody later detected was not tested at this time.

The authors refer to focal epilepsy with “clonus in her left upper limb”. This should likely be “clonic movements” rather than “clonus” which is not generally epileptic.

It would be helpful to provide further data from the 2nd CSF – WBC, protein, glucose, IgG index, OCB.

Author Response

The abstract says the patient had “transient” aphasia, while the body of the case report states the patient had “progressive” aphasia. These are different and it should be clarified whether aphasia was transient or progressive

The abstract was rephrased: “Three months after ITP, the patient presented progressive aphasia that recovered spontaneously, fluctuating cognitive deficits and focal epilepsy responsive to levetiracetam”.

Please specify in line 78-79 antibodies to onconeural antigens that were tested and whether this was on serum or CSF.

Done (line 79-80)

 Please clarify that the antibody later detected was not tested at this time.

Done (line 108)

 The authors refer to focal epilepsy with “clonus in her left upper limb”. This should likely be “clonic movements” rather than “clonus” which is not generally epileptic

Done (line 98)

It would be helpful to provide further data from the 2nd CSF – WBC, protein, glucose, IgG index, OCB.

Done (line 106-107).
